# Evaluation Beyond Task Performance: Analyzing Concepts in AlphaZero in Hex

**Charles Lovering**\* **Jessica Zosa Forde**\*
**George Konidaris** **Ellie Pavlick** **Michael L. Littman**
Department of Computer Science
Brown University
`{first}_{last}@brown.edu`

## Abstract

AlphaZero, an approach to reinforcement learning that couples neural networks and Monte Carlo tree search (MCTS), has produced state-of-the-art strategies for traditional board games like chess, Go, shogi, and Hex. While researchers and game commentators have suggested that AlphaZero uses concepts that humans consider important, it is unclear how these concepts are captured in the network. We investigate AlphaZero's internal representations in the game of Hex using two evaluation techniques from natural language processing (NLP): model probing and behavioral tests. In doing so, we introduce new evaluation tools to the RL community, and illustrate how evaluations other than task performance can be used to provide a more complete picture of a model's strengths and weaknesses. Our analyses in the game of Hex reveal interesting patterns and generate some testable hypotheses about how such models learn in general. For example, we find that MCTS discovers concepts before the neural network learns to encode them. We also find that concepts related to short-term end-game planning are best encoded in the final layers of the model, whereas concepts related to long-term planning are encoded in the middle layers of the model.

## 1 Introduction

AlphaZero [55], a reinforcement-learning agent that combines Monte Carlo tree search [7] with deep reinforcement learning, has achieved impressive performance at games like Go, chess, shogi, and Hex. Domain experts have observed that AlphaZero uses, but does not master, identifiable game concepts. For example, despite being exceptionally strong overall, AlphaZero appeared unable fully to project the implications of a *ladder*—an important concept in the game of Go [60].

Good performance can mask flaws in deep learning systems generally [2, 49, 22, 9, 14, 69, 10], and deep reinforcement-learning systems in particular [66, 70]. Evaluating systems in terms of task performance alone makes it impossible to know if systems are "right for the right reasons" and difficult to predict how they will generalize to new situations [67, 68]. Recently, the field of natural language processing (NLP) has begun to develop evaluation techniques that go beyond "just" task performance [3, 4]. For example, some techniques, *probing classifiers*, inspect the form of models' internal representations to test whether they are consistent with linguistic theory [1, 8, 58]; other techniques, *behavioral tests* or *challenge sets*, evaluate specific types of out-of-distribution generalization to assess whether models encode human-like inductive biases [13, 19, 32, 34, 36, 41, 46, 63]. Although these techniques are only used over known concepts, often requiring expert domain knowledge to define, the insights generated by these techniques are not only scientifically interesting, but have begun to yield actionable insights on how to employ and improve models. For example, Geva et al.

---

\*Equal contribution.

36th Conference on Neural Information Processing Systems (NeurIPS 2022).

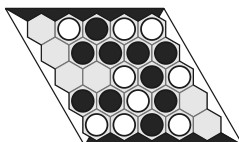 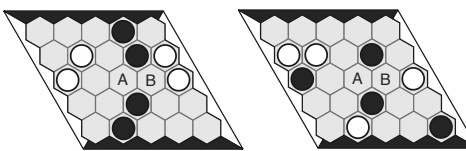

(a) **An example winning board for black**, which connects the black edges together.

(b) **Short- vs Long-term concepts.** Left: If black plays A or B, black immediately wins; the board concept containing A and B, *bridge* (See 2a), is relevant in the *short-term*. Right: A and B can help black win only in the *long-term*.

Figure 1: To win in Hex, a player must use their pieces to form a connecting chain between the edges matching their color (a). Hex boards can be of varying size; we evaluate AlphaZero on a 9x9 board. In our work, we make a distinction between short- and long-term concepts (b).

[20] found that multilayer perceptrons (MLPs) in transformer layers act as key-value memories, and Meng et al. [43] leveraged this understanding to manipulate model behavior in a controlled manner.

We demonstrate how these ideas leveraged in NLP can be applied to reinforcement learning, testing the capabilities and limitations of our models. Specifically, we leverage the above-described analysis techniques—probing classifiers and behavioral tests—to interpret AlphaZero's behavior at a conceptual level. We use probing classifiers to determine if reinforcement-learning agents encode tactical and strategic conceptual information. However, probing performance alone is insufficient: Information may be encoded but not used [37]. To address this issue, we also use behavioral tests, which evaluate an agent's decisions in a situation tailored to require the understanding of a specific concept.

Given these concept-level evaluation methods, we do an in-depth study of AlphaZero (AZ) trained to play Hex. Hex is a board game similar to Go (§2), but provides an opportune test bed for analysis: the game is complex, yet has perfect-play baselines for smaller board sizes. Moreover, concepts and strategies within Hex have been studied by the Hex-playing community, making Hex a rich vector for study. Furthermore, the clear behavioral expectations in Hex make it easier to interpret and guide future work.[2] In our work, we probe for concepts that are taught to new players of Hex, like the *bridge* (Fig. 2a), and test that the model is able to use them to win games. We investigate how concepts are represented within AZ, when concepts are learned during training, and where concepts are represented within AZ's neural network.

Overall, our findings suggest that there are some regular dynamics to how AZ learns concepts with the game of Hex, and generate interesting predictions about how AZ behaves in general, which could be tested in other games. (In fact, concurrent work on chess [42] and Go [61] already begin to provide convergent evidence that some trends we see here generalize elsewhere.) We introduce a novel way of analyzing deep RL models on which such subsequent work can easily build.

In summary, our main contributions are:

1. We adapt several evaluation techniques from NLP to the RL setting, and illustrate how evaluations other than task performance can be used to provide a more complete picture of a model's strengths and weaknesses.

2. We analyze a top-performing model from Jones [30], and find that (1) short-term end-game concepts are best represented in the final layers of the network, whereas long-term concepts are best represented in the middle layers of the network; (2) concepts appear to originate with MCTS—with MCTS overriding the deep learning policy prediction early in training—but later in training these concepts are incorporated directly into the model's network.

---

[2]Again drawing inspiration from work in NLP, Linzen et al. [35] focused on one syntactic phenomenon in simple current neural networks, and the basic insight and methods enabled later studies to make broader claims about learning linguistic structure in many types of models [26, 63, 64].

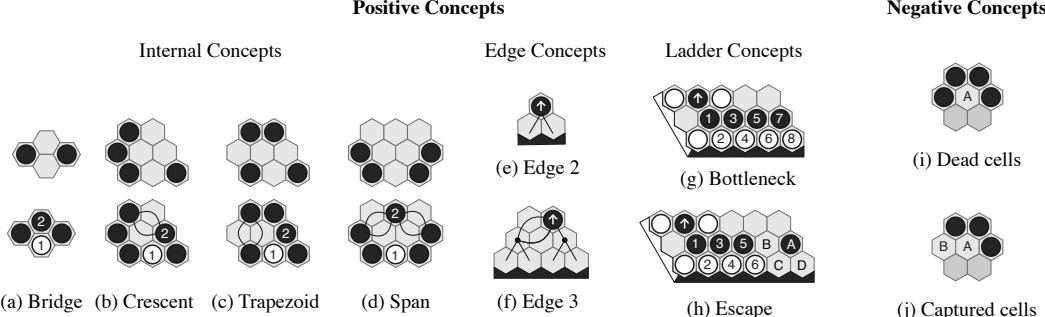

**Positive Concepts**

Internal Concepts      Edge Concepts    Ladder Concepts

**Negative Concepts**

(e) Edge 2

(g) Bottleneck

(i) Dead cells

(a) Bridge  (b) Crescent  (c) Trapezoid  (d) Span     (f) Edge 3     (h) Escape     (j) Captured cells

Figure 2: **Hex templates exemplifying game concepts.** Concepts within the game of Hex are templates on the board formed by a player's pieces with known strategic and tactical implications. Positive concepts provide the player with the concept with multiple ways to connect the pieces within the concept together, despite possible attacks from the opponent. An example of such a concept is the bridge (a). If white plays move 1, black can connect the two pieces of the bridge by playing move 2. Negative concepts change the strategic value of specific open spots of the board, such that the opponent is disincentivized to play those open spots, such as move A in (i). Each concept is further described in our Supplementary Material. Arrows indicate that the piece is connected to the opposite side of the board; the lines show the bridge concept within the other concepts.

## 2 Concepts in Hex

We study AlphaZero (AZ) agents trained to play Hex [17], a game where two players take turns filling cells until one player builds a chain across the board. Hex has well-studied rules, reasonable computational costs, and it can be evaluated against perfect play, making it an ideal experimental vehicle for model probing. Unlike Go, there are no captures; once a cell is filled with a piece, the pieces stays there for the remainder of the game. For example, in Fig. 1a, white must connect pieces from left to right (marked on the edges), and black top to bottom. Hex cannot end in a tie [15], and given perfect play, black, the first player, will win [17]. [3]

In Hex, certain templates – patterns of cells – have been recognized as useful. Because building up a chain from one side of the board to the other is easily thwarted, a key part of learning how to play Hex is recognizing when it is possible to connect groups of pieces together. We consider these templates to be *concepts* within the game. Concepts in Hex have different move implications depending on the condition of the board. We discuss two conditions, long- and short-term, in detail and provide additional discussion of conditions in the Supplementary Material.

While the properties of concepts are debated [39], here, in a board game setting, we consider a concept to be a useful template that generalizes across board configurations. For our purposes, "understanding" a concept amounts to recognizing it and leveraging its implications during gameplay (§2.2).

### 2.1 Long-term vs short-term Concepts

We define a concept in Hex to be *short-term* if its use is sufficient to win the game. Concepts are typically short-term when there exists a connection between the concept and the player's board edges. For example, in Fig. 1b, moves A and B belong to the *bridge* concept (see Fig. 2a). In the left board, the board edges are connected, and all that is required for black to win is to play moves A or B. Conversely, *long-term* concepts are insufficient to winning the game if immediately used. In the right board of Fig. 1b, playing the *bridge* concept is useful but insufficient for winning immediately; the concept is not connected to the edges of the board. Empirically, we find that whether the concept is short- or long-term has a significant impact on AlphaZero's representations (Fig. 4).

---

[3]Hex is often played with a "swap rule" that makes the game more even between black and white. See Jones [30], whose implementation we use, for further discussion on the swap rule. Jones did not include it to simplify the game implementation.

## 2.2 Concept Taxonomy

From Seymour [54] and King [33], we identify the nine concepts that we use in our analysis, summarized in Fig. 2. These concepts fall into four categories, as described below.

**Positive Concepts** With the goal of Hex being to build a chain across the board, it is helpful to recognize when cells are virtually connected, that is, even in response to perfect adversarial play, the cells are guaranteed to connect [23, 47]. There are a few different types of positive concepts. All the positive concepts favor the player that owns the concept on the board.

Internal concepts are templates that appear within the interior of the board. The *bridge* (Fig. 2(a)) is the simplest such concept. The larger internal templates – *crescent*, *trapezoid*, *span* (Fig. 2(b,c,d)) – provide several possibilities to connect a player's pieces. *Edge* concepts concern connecting a single cell to a given edge. Ladders in Hex are similar to ladders in Go. There are two different ladder concepts, *bottleneck* (g) and *escapes* (h). A bottleneck favors the defender, because the attacking player cannot break through. An escape, however, allows the attacker to break through.

**Negative Concepts** Negative concepts do not inform one on which actions to play, but rather, which actions not to play. *Dead cells* (Fig. 2(i)) cannot impact the outcome of the game regardless of the color with which they are filled. While it is in general difficult to compute if a cell is dead [6], there are some known templates where it is easier to deduce. If a player can make a cell dead, that cell is *captured*. Both captured and dead cells should not be filled.

## 3 Designing Probing Tasks and Behavioral Tests for Hex

To understand the concepts encoded within AlphaZero, we probe its internal representations; to evaluate if these concepts are used by AZ, we test its behavior on tailored board configurations. By evaluating AZ across training checkpoints, and across neural network layers, we can build up an understanding of how and where the model recognizes these concepts. Specifically, we evaluate the top-performing agent trained by Jones [30] across 21 training checkpoints. We additionally evaluate other publicly available agents [30] from that differ by width and depth and we report on those results in the Supplementary Material. Code from Jones is available under the MIT License.

### 3.1 Reproducibility

Our code and results are publicly available [38]. Furthermore, we release example images of boards created for our probing classifiers and videos of the behavioral tests. The code, results and examples can be found at `https://bit.ly/alphatology`. Our repository is also available on GitHub at `https://github.com/jzf2101/alphatology`. We report hyperparameters in the Supplementary Material. All error bars presented in plots are one standard deviation above and below the mean. We used NVIDIA GeForce RTX 3090. The total compute across all experiments was about 24 GPU hours. We present a breakdown of the compute in the Supplementary Materials.

### 3.2 Representational Probing

Model probing measures how well a model's learned representations encode a known concept [1]. To make a model probe, one labels examples, such as Hex boards, with the presence or absence of a concept, such as the bridge concept (see Figure Fig. 2). Next, one collects the model's activations for each example, and then trains a linear classifier (the *probe*) to predict the presence of the concept based on activations. The linear classifier's test performance is used to interpret how well the original model encoded the concept.

We train a linear probe for each concept over each layer of AZ's network body. We follow a procedure similar to Tenney et al. [58]: $H^{(0)}$ is the state of the board that is used as input into AZ. For each board, we record the label $y_k$ to indicate the presence or absence of the concept $k$ on the board. $H^{(l)}, l \in 1..L$ is the activation of the layer $l$ of AZ's network body.[4] We then train linear classifiers $\mathcal{P}^{(l)}, l \in 0..L$ per layer to predict the presence vs. absence of a concept in a given board. These classifiers are our concept probes.

---

[4]We never use activations from multiple layers at once, layer weights , nor AZ's value/policy outputs.

It is important to compare probing results against a baseline. We follow Hewitt and Liang [24]'s procedure to measure *concept selectivity*, the delta between probing accuracy over a concept and random control. To form the random control, for each board $H^{(0)}$ in the probing dataset, we map each cell in that board to a random cell in a consistent manner to form the transposed board, $H_s^{(0)}$. In this way, the same information is encoded in the original boards, but we expect the shuffled boards to be meaningless in Hex. Next, we train a set of linear probes $\mathcal{P}_s^{(l)}, l \in 0..L$ over the control boards $H_s^{(0)}$ to predict $y$. Now, finally, we can compute the concept selectivity by finding the delta in accuracy between $\mathcal{P}_s^{(l)}$ and $\mathcal{P}^{(l)}$. Concept selectivity is the performance of a probe beyond the performance of a probe on a control task, adding context for interpreting the results.

**Implementation Details.** Each concept is defined by a set of boards with vs. without that concept. We train and evaluate probing classifiers over AlphaZero's encoding of these boards. To generate a set of ($N = 2500$) boards for each concept, we translate the minimal templates across an empty board. Then, we add random enemy pieces to the board. Negative instances of a given concept match the statistics of the positive examples, except that the pieces pertaining to the concept template are randomly moved across the board. This constitutes the long-term version of a concept. To form the short-term version of a concept, we connect the template to the edges of the board.

## 3.3 Behavioral Tests

Where model probing asks if concepts are represented within the model, behavioral tests asks if the model knows how to use the concept in gameplay. To interpret the behavioral tests, they must have clear behavioral expectations. We construct our behavioral tests for positive concepts (§2) to be forced: If AZ understands the concept and plays the expected moves, AZ will win and pass the test; otherwise, AZ will lose the game and fail the test. For negative concepts (§2), we have clear behavioral expectations. Dead and captured cells should never be filled. Thus, the behavioral test for negative cells checks that during a selfplay continuation of a board containing a dead (or captured) cell, the agent does not fill that cell.

Success on these behavioral tests are necessary but are alone insufficient to establish that the model has the concept. A negative result means that AZ is unable to use the concept, whereas a positive result means that AZ can use the concept to win games in forced situations. This approach has been used to test large language models. Specifically, we are inspired by Ettingers [13, 46] who uses simplified language to ask if the model is able to use a concept like negation or syntax when needed. If the model still fails to use the concept, then this good evidence that the concept is not represented.

**Implementation Details.** For each concept, we create behavioral tests that comprise a board, forcing moves, and expected moves. (For the dead and captured cells there are no forcing or expected moves, only the moves to avoid). We discuss the motivation for our behavioral test setup and its connection to AlphaZero's style of gameplay in the Supplementary Material. By way of example, see Fig. 3. To generate a set ($N = 100$ samples) of boards, we translate the templates to a sampled valid board position. We then connect the concept to the edges of the board. Finally, we add connections for the defending player up to the region of the concept *such that if the attacking player fails to complete the behavioral test, the other play would win*. Finally, we add the appropriate number of random pieces such that the board position is valid. The behavioral tests determine when/if AZ learns to navigate these situations correctly.

## 3.4 Considerations

**Limitations.** While we find a consistent relationship between the probing and behavioral tests in Section 4.3, we do not run a counterfactual study. For example, we do not show that mistakes in recognizing a concept on a given board, lead to mistakes in using that concept. Studying the causal mechanisms of how the concept representations detected by model probing impact downstream model behavior is a rich direction for future work.

In our behavioral tests, all the concepts we tested share the property that they are about to be connected. (They share this property because we connect the concept so that the behavioral expectations are clear, Fig. 3.) All the concepts being tested could be interpreted as "interrupt the opponent's soon-to-be-winning chain." However, the behavioral tests of different concepts report different performance levels

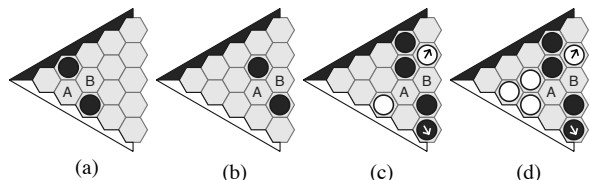

(a)        (b)        (c)        (d)

Figure 3: **Creating behavioral tests from concept templates.** To evaluate AlphaZero's ability to utilize concepts during game play, we build synthetic boards where utilizing the strategic advantages of the given concepts allows the player to win the game. In this example, we demonstrate the building of a behavioral test for the bridge concept (Figure 2a). The minimal template (a) is translated to a random position on the board (b). Then both players' pieces are connected to their respective edges they need to utilize to win the game (c). Finally, the minimum number of noise pieces necessary to form a valid board are added (d). Cells A, B are used to define the behavioral test. If white plays A, black must play B to win the game. (Only half a 5x5 board is shown for space.)

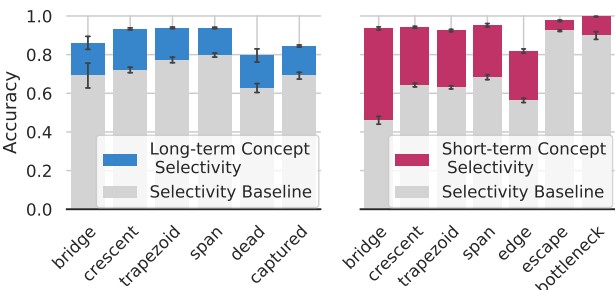
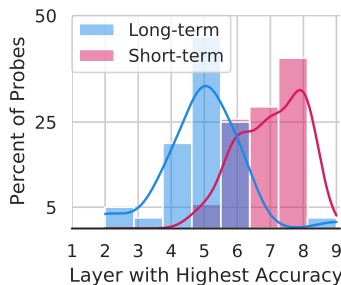

(a) **AlphaZero successfully encodes long-term and short-term concepts.** The selectivity [24], indicated by the colored bars, are the accuracy of a probe trained to identify a concept based on network activations, minus the accuracy of a selectivity baseline. The selectivity baseline randomly maps board pieces so that the new boards do not contain structures known to be relevant to Hex. We report selectivity based on the layer with the highest test accuracy.

(b) **Long-term concepts are best represented in the middle layers of the network whereas short-term concepts are best represented in the final layers of the network.** Each distribution shows the layer in which probes had the highest accuracies.

Figure 4: **Probing performance on long- and short-term concepts.**

(though admittedly similar), suggesting that the differences of concept are still relevant. Furthermore, the agent would still have had to use the targeted concepts as a constituent of "interrupt the opponent's soon-to-be-winning chain."

**Societal Impact.** Deep reinforcement learning models such as AZ are not interpretable [12], and yet are being applied to impactful, real world domains [21, 50, 18, 5, 31, 44]. In this work, we start to analyze how AZ comes to its decisions. Understanding a model's decisions is critical for accountability, but it does open the door to some avenues for exploitation. If it is known that a given algorithm does not reason about a concept well, this could be leveraged for ill. However, this risk only strengthens the argument for uncovering such problems and fixing them to prevent such risks.

# 4 Results

To understand which concepts AlphaZero (AZ) learns, we examine if its neural network activations encode the concepts (probing tests) and if AZ can use the concepts to win games (behavioral tests).

## 4.1 AlphaZero Recognizes and Uses Concepts

AZ successfully encodes short-term concepts, with high selectivity scores (Fig. 4a). The long-term concept scores are also learned, but with slightly lower scores. By the end of training AZ is able to use all the positive concepts to win games (Fig. 5a).

AZ improves on the behavioral tests 50% of the way through training. Unsurprisingly, the MCTS passing rates increase before the policy network passing rates (Fig. 5), though it need not have been the case. It is possible for AZ to represent the concepts before MCTS used them – possibly via the signal through the value prediction.

Interestingly, the relative magnitude of the correct action logits initially increases earlier. To specify how we measure this, we need to cover two definitions. First, the logits are the outputs of the modules (MCTS or policy prediction head). Second, we use a Z-score, which reports the number of standard deviations higher a given value is than the mean of the population. In our case, the blue line in Fig. 5a, reports the proportion of cases where the Z-score of the correct action is greater than 1. So, this value captures when the correct action becomes more likely throughout training.

The trend in Fig. 5a suggests "pre-conceptual" information is learned, and coalesces (for bridge) 60% of the way through training into an actionable understanding of the concept. (In Section 4.4, we investigate further and find that this "pre-conceptual" information is not board structure.)

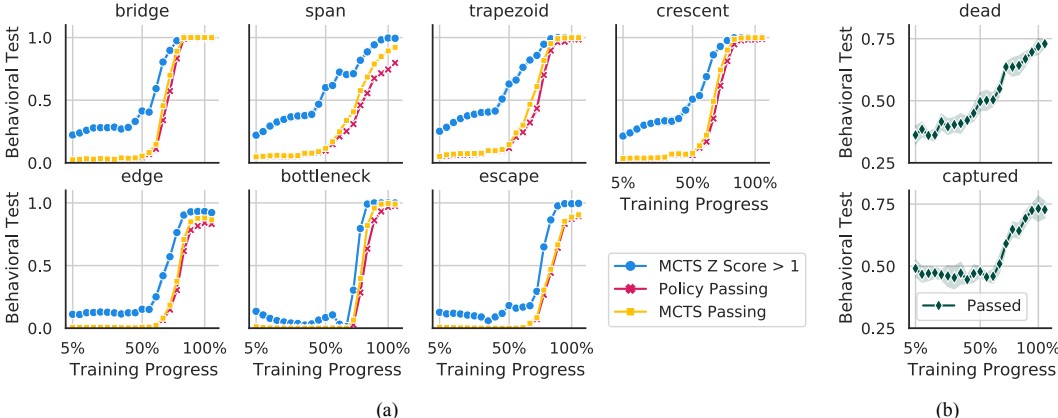

Figure 5: **(a): AlphaZero learns to use positive concepts.** At each checkpoint, we present AlphaZero (AZ) with a set of example boards that test its ability to utilize each concept. MCTS and the policy network both select actions that pass our behavioral tests with increasing frequency throughout training. We additionally report the rate at which the logit score of the action that passes our behavioral test is one standard deviation above the mean logit score (z score > 1).
**(b): AZ does not fully use the negative concepts.** Passed denotes the rate at which AZ avoids the negative concept throughout selfplay rollouts. So, at the end of training, AZ plays moves in 25% of our behavioral tests that will not impact the game [6].

AZ also improves behaviorally upon negative concepts. However, it does not reach a perfect passing rate (Fig. 5b). The probing performance for the negative concepts, shown in Fig. 4a, is also lower than for other concepts. This aligns with evidence that AZ wastes moves at the end of the game. This highlights a weakness in AlphaZero and a risk: Some concepts may be "provable" and useful to people, but "deemed" less important by AZ – an agent that plays remarkably well. We will have to consider this dynamic as people begin to try to learn concepts from machines.

## 4.2 Long-term and Short-term Concepts are Best Represented in Different Layers

Fig. 4b highlight that short and long-term concepts are best represented at different layers. Long-term concepts, by the end of training, are best represented in the middle layers of the network. Short-term concepts, throughout training, are best represented in the upper layers of the network. We have two conjectures, which are not mutually exclusive, for why this may be the case. We leave the verification/refutation to future work. (1) Short-term concepts generally require more global information and so require more depth; (2) Short-term concepts factor directly into action selection and so are more proximal to the final task-specific layer of the network.

We find that two and four layer networks demonstrate the same pattern; see our Supplementary Material. Lastly, McGrath et al. [42]'s results in chess are similar, although they did not directly study short- vs long-term concepts. The following is our interpretation of the results shown in their plots: "in check" (short-term) was best represented higher in the network and "material imbalance"

(long-term) was largely represented lower in the network. (Informally categorized, the short-term concepts (McGrath et al.'s Fig. 2)[c,e,f] are better represented in higher layers than are the long-term concepts (Fig. 2)[a,g,h,i].)

## 4.3 Improvements in Behavioral Tests Occur Before Improvements in Probing Accuracy

Where the network body processes board configurations, MCTS directly governs the decision-making procecess. In principle, either module could be the first to discover the game concepts. For instance, the network body could start to represent concepts via updates to the value function, later enabling MCTS to successfully navigate the concept templates. We find that the first improvements in behavioral tests precede the first improvements in probing accuracy (Fig. 6), where we consider first improvement as the epoch with a 5% increase over the baseline value. MCTS seems to discover concepts, especially the internal concepts. Then, as the policy network is trained to match the MCTS logits, the concept representation is absorbed into the network.

We find evidence that the structure of the board is first learned at about the same time other concepts are learned. In Section 4.4, we discuss how we probe for this structure, determining if AZ encodes the relative distance between all cells on the board. The last column of Fig. 6, labeled *structural*, shows that improvements on this concept occur at a similar time as to other concepts. This suggests that AZ does not learn concepts according to an obvious order or curriculum, but rather, concepts of differing levels of complexity develop in parallel.

The results shown in Fig. 6 outline the relationship between the concepts as they are learned during training, but does not establish a causal relationship between how the concept representation (as detected by the probing models) impacts how the model uses that concept.

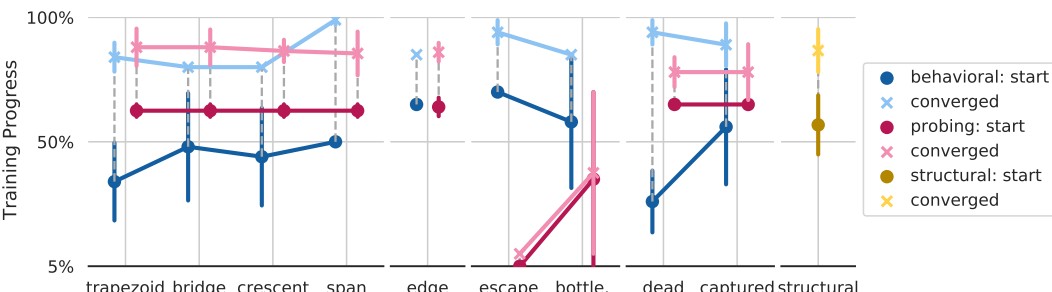

Figure 6: **Improvements in behavioral tests occur before improvements in probing accuracy.** Each point marks the mean checkpoint in training that AZ started to learn (or converge upon) the behavioral (or probing) test. Error bars are standard deviation across random seeds and short- vs long-term. We group the concepts as in Fig. 2. While behavioral tests start to improve before the probing, they both converge near the end of training. Exception: the ladder escape and bottleneck concepts (Fig. 2h) are easy for the probes to detect, consequently having low selectivity (Fig. 4a), perhaps because they occur along the edges of the board, and as a result, have fewer possible configurations. The concept structural, (§4.4), which evaluates how well AZ's cell embeddings capture Hex's neighborhood structure, is learned and converges in a similar time frame as the probing task.

### 4.4 AlphaZero's Cell Embeddings Capture the Structure of the Board

Understanding Hex's concepts requires under-
standing the board's structure, that is, which
cells connect to which other cells. AlphaZero
(AZ), with its feed forward network architec-
ture, does not *a priori* represent this struc-
ture. Fig. 5 suggests that some information
is learned before the model is able to use the
concepts. A possibility is that AZ spends the
initial portions of training building up a repre-
sentation of the board. However, we find no
evidence that the neighborhood structure is
learned in the initial stages of training, rather,
it appears to be learned only after the first
improvements in behavioral tests (Fig. 6).

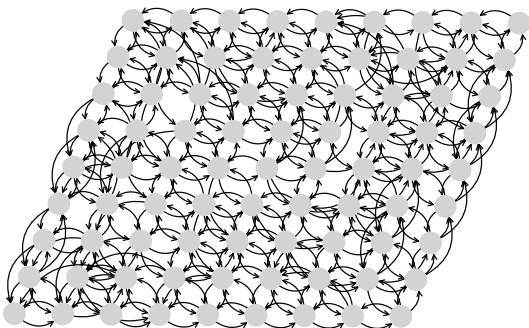

Figure 7: **Implicit board structure.** Arrows mark
the learned nearest neighbors of each cell.

To study if Hex's board is represented in AZ, we hypothesized that the structure of Hex's board is
implicitly learned by AZ's first layer. For each Hex cell, we extract a cell embedding from the first
layer of AZ.[5] Next, we compute the dot-products between each cell embedding. The dot-product
score between ground-truth neighbors increases throughout AZ's training.

The nearest-neighbors (according to the dot-product scores) nearly match the ground-truth by the 15th
of 20 checkpoints, and eventually match the ground-truth before deviating slightly. We evaluate how
well the dot product scores align with the ground-truth cell distances via a ranking metric, Normalized
Discounted Cumulative Gain (NDCG). The NDCG first improves about 50% of the way through
training as highlighted in Fig. 6. Details on the learned structure are in the Supplementary Material.

## 5 Related Work

**Applying MCTS and Deep Learning to the Game of Hex.** Jones [30], whose trained agents we
use, studied scaling laws between various parameters, finding a mathematical relationship between
compute, board size, and agent performance. This type of result can help others make informed
tradeoffs when training board-game-playing agents. Where Gao et al. [16] and Takada et al. [57]
both use value functions in concert with MCTS to play Hex, MoHex [27] combines programmatic
connection detection, pattern matching, and MCTS, attacking Hex for the 9×9 grid.

**AlphaZero, anecdotally, uses concepts, but only achieves a partial mastery.** Domain experts
have observed that AlphaZero [55] and related agents that combine Monte Carlo tree search [7]
with deep reinforcement learning use identifiable concepts within board games. Michael Redmond
identifies several common gameplay concepts demonstrated by AlphaZero [11] in AlphaZero's Go
matches against Lee Sedol. Silver et al. [56] noted that common human-played corner moves in
Go—*joseki*—were used by AlphaZero during self-play training before being abandoned, presumably
as other more effective moves were found. Additional analysis provided by Tian et al. [60] note that
Elf OpenGo only partially mastered ladder sequences within the game. Chess commentator Antonio
Radić [51] detailed how AlphaZero used *zugzwang* [65] in the course of defeating Stockfish [52].
Experts have already incorporated some of AlphaZero's innovations into their play [45, 53].

**Probing neural networks for human concepts.** A range of linguistic concepts have been detected
in NLP models using probing models [8, 48, 40, 59, 25]. For example, Tenney et al. [59], with
probing models, found that BERT appears to best encode linguistic concepts in the same order those
concepts might be processed by the typical NLP pipeline. There has been some discussion on what
good probing accuracy signifies. Hewitt and Liang [24] calls for baseline controls, and to measure
only the gain in accuracy compared to these baselines. Voita and Titov [62] found that measures

---

[5]AZ's first layer takes a flattened 1-hot encoding of the board and matrix multiplies it with a weight matrix.
The embedding of each cell can be spliced out of that weight matrix. We only extract the cell embeddings for
the current player. This constitutes a structural probe [25]. A structural probe tests the relationship between
neural network activations (or weights).

beyond accuracy, namely Minimum Description Length (MDL)—which is used to measure how easy it to detect a given concept—provided more stable results.

**Understanding reinforcement-learning agents trained to play board games.** Sadler and Regan [53] cover how some particulars from AZ's gameplay has impacted high-level chess. Concurrent to our work, McGrath et al. [42] also looked at how AZ acquires game knowledge (which we term *concepts*) in chess. Using similar probing techniques, and the original (and larger) AZ, they find that it learns to encode many of the prototypical chess concepts. Related, but not focused on model understanding per se, Jhamtani et al. [28] collected an annotated set of chess games. This type of resource is similar to what is used by McGrath et al. [42] for their probing task. In our work, we focus on controlled programmatic generation of boards with (and without) concepts present. While these board instances may not occur during selfplay, the human annotated games are also off-distribution with respect to AZ. Beyond finding similar results in a different game (Hex vs chess), we also leveraged behavioral tests, tying the concept representation to agent behavior.

## 6 Discussion

Our analyses suggest that AlphaZero (AZ) learned to both represent and use concepts that humans consider important when playing Hex. However, we found that AZ is sometimes either blind or agnostic to a winning move. This had interesting ramifications. For instance, the negative concepts, especially dead cells, were not as well-encoded or used compared to other concepts. This may be because AZ has no "qualms" with playing wasted moves, like dead cells, if doing so doesn't change the outcome of the game.

A layerwise analysis showed that the same concept is represented most strongly in different layers, depending on its context: short-term concepts that inform actions at the end of the game are encoded in the upper layers of the model, whereas long-term concepts are absorbed deeper into the network. This absorption mirrors how AZ's policy head was trained to predict the policy outputs of MCTS.

Combining both representational and behavioral approaches to analyze reinforcement-learning agents allows for a fuller understanding of how agents learn. Studying the representations of concepts allows us to ask (and answer) a rich set of questions about where those concepts reside. Studying the behavior of the model with respect to a given concept tests that this representation can be translated into action. Behavioral tests can also determine whether the model may be using known heuristics. They are complementary approaches. In future work, applying a causal approach to the study of agents' policies will further illustrate how well these agents understand these concepts.

## Acknowledgements

We would like to thank the reviewers across different iterations of this work; they helped us clarify our work and findings. This research was conducted using computational resources and services at the Center for Computation and Visualization, Brown University. Charles Lovering was supported in part by the DARPA GAILA program. This work received support from the ONR PERISCOPE MURI award N00014-17-1-2699.

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
