# Supplementary Materials
## Evaluation beyond Task Performance:
## Analyzing Concepts in AlphaZero playing Hex

**Charles Lovering**[*]  **Jessica Zosa Forde**[*]
**George Konidaris**   **Ellie Pavlick**   **Michael L. Littman**
Department of Computer Science
Brown University
{first}_{last}@brown.edu

Appendix A reports implementation details, hyperparameters and compute requirements.
Appendix B gives more details on each concept introduced in the main body of the paper.
Appendix C demonstrates how AlphaZero often wastes moves.
Appendix D has additional results across the different architectures.

## A   Implementation Details

We use agents trained by Jones [5]. See Table 1 for hyperparameters and relative agent strengths. We used NVIDIA GeForce RTX 3090, to generate boards and encode them. The compute is reported in Table 2 For all results in the main body of the paper we use the model `grubby`. We report additional results with other models below in Appendix D. The code, results and examples can be found at `https://bit.ly/alphatology`.

---

[*]Equal contribution.

36th Conference on Neural Information Processing Systems (NeurIPS 2022).

| agent run | Jones [5] full agent run name | depth | width | MCTS nodes | train ckpts | win rate against MoHex as black | Elo |
|---|---|---|---|---|---|---|---|
| `grubby` | **2021-02-20 22-35-41 grubby-wrench** | **8 layers** | **512 neurons** | **64** | **20** | **0.922** | **-0.345** |
| `recent` | 2021-02-20 21-33-42 recent-annex | 8 layers | 256 neurons | 64 | 19 | 0.77 | -0.361 |
| `baggy` | 2021-02-20 22-18-43 baggy-cans | 4 layers | 512 neurons | 64 | 20 | 0.922 | -0.388 |
| `vital` | 2021-02-20 22-55-43 vital-bubble | 2 layers | 1024 neurons | 64 | 20 | 0.922 | -0.400 |

Table 1: **Agent hyperparameters.** The win rate is the agent's win rate *as black* vs MoHex [4] *without the swap rule*. Under perfect play in Hex, black cannot lose. The Elo rating for each agent is calculated based on trials against MoHex and against the subset of agent runs Jones [5] evaluated against against MoHex. Because Jones [5] fixes the number of MCTS nodes used to compare against MoHex at 64, we do not consider alternate agent configurations that expand or contract the number of nodes when calculating the Elo rating.

| script | time | concepts | seeds | parallel | total |
|---|---|---|---|---|---|
| `encode` | $8.34 \pm 4$ m | x9 | x3 | - | 216 m |
| `probing` | $21.75 \pm 16.35$ m | x9 | x3 | /4 | 146 m |
| `positive` | $14.31 \pm 8.23$ m | x7 | x3 | /4 | 75 m |
| `negative` | $23.48 \pm 5.46$ m | x2 | x3 | /4 | 35 m |

Table 2: **Compute.** For `grubby` model, the total is about 8GPU hours. The other models take less time than this model. The total run time is about 24GPU hours. We used NVIDIA GeForce RTX 3090.

| experiment | parameter | $N$ |
|---|---|---|
| probing | training examples | O(2000) |
| | test examples | O(500) |
| | seeds | 3 |
| behavioral tests | examples | 100 |
| | seeds | 3 |

Table 3: **Experiment hyperparameters.**

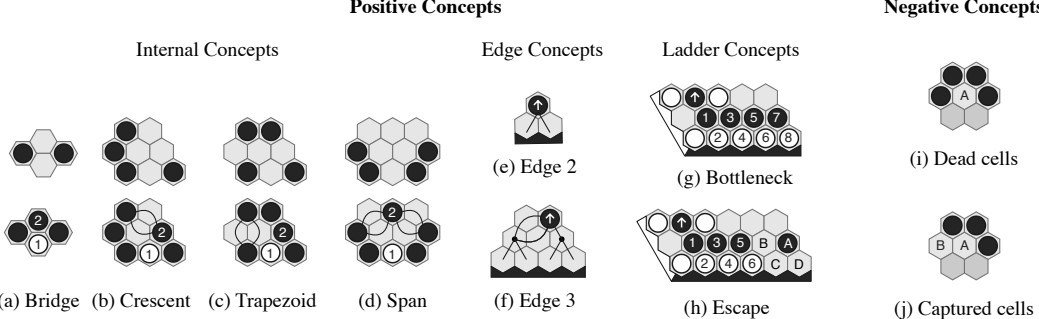

**Positive Concepts**     **Negative Concepts**

Internal Concepts     Edge Concepts   Ladder Concepts

(e) Edge 2     (i) Dead cells

(g) Bottleneck

(a) Bridge  (b) Crescent  (c) Trapezoid  (d) Span     (f) Edge 3     (h) Escape     (j) Captured cells

Figure 1: **Hex templates exemplifying game concepts.** Concepts within the game of Hex are templates on the board formed by a player's pieces with known strategic and tactical implications. Positive concepts provide the player with the concept with multiple ways to connect the pieces within the concept together, despite possible attacks from the opponent. An example of these properties is the bridge (a). If white plays move 1, black can connect the two pieces of the bridge by playing move 2. Negative concepts change the strategic value of specific open spots of the board, such that the opponent is disincentived to play those open spots, such as move A in (i). Each concept is further described in our Supplementary Material. Arrows indicate that the piece is connected to the opposite side of the board; the lines show the bridge concept within the other concepts.

## B   Concepts in Hex

**Internal Concepts**   With the goal of Hex being to build a chain across the board, it is helpful to recognize when cells are virtually connected, that is, even in response to perfect adversarial play, the cells are guaranteed to connect [3, 9]. The bridge (Fig. 1a) is the simplest such concept. The larger internal templates – crescent, trapezoid, span (Fig. 1(b,c,d)) – provide several possibilities to connect a player's pieces.

**Edge Concepts**   An edge template guarantees a connection from a single cell to a given edge [6, 11]. Recognizing these edge templates is important for building effective strategies. In this work, we consider the edge concept to be the two templates shown in Fig. 1(e, f). In Fig. 1(e, f), black can connect to the bottom wall irrespective of how white attempts to sever the connection.

**Ladder Concepts**   Ladders are common in Hex. While similar in spirit to ladders in Go, there are some technical differences. We analyze two different ladder concepts. *Bottlenecks* are a defensive concept that leads to the ladder. The defender holds off the attacker (in Fig. 1g white successfully defends against black). If the attacking player (here black) continues the ladder, the defending player (here white) must block each move or lose the game. However, the defending player will eventually win, as each defensive move builds up a chain across the board. Ladder *escapes* have a different outcome. If there is already a cell in path of the ladder, like A in Fig. 1h, then the attacking player will win. When black plays B connecting to A, white cannot defend against both C and D.

**Dead Cells (Useless Triangles).**   Some empty cells cannot impact the outcome of the game regardless of with which color they are filled. A cell being dead is a global fact of the board, and is difficult in general to compute [2]. However, there are known templates where it is relatively easy to deduce that a cell is dead, and it is these templates with which we test the model. Among the simplest is the "useless triangle", which is the precursor of the more general notion of dead cells (Fig. 1e). If white plays A, she does not restrict black's territory. To do so she would have to play both other empty cells, after which her move into A would not have any use. For black, playing A doesn't hinder white, nor provide any new territory.

**Captured Cells.**   An empty cell is considered captured when it is effectively filled by a player. The templates have (at least) two empty cells A and B (Fig. 1f). If the cells are black-captured, it means that if white intrudes into the template (playing A), then black can respond by playing B and making A dead.

# C  Characterizing AlphaZero's Gameplay of Hex

AlphaZero (AZ) uses both deep learning and MCTS. The deep network, $f_\theta(s)$, takes in boardgame input, $s$, and produces two forms of output: the value estimate of the board, $v$, and a move prior, $\mathbf{p}$. These prior probabilities are then used for MCTS, which outputs the MCTS probability distribution over actions, $\boldsymbol{\pi}$. In the game of Hex, rewards, $z$, and value estimates are scaled within $[-1, 1]$ and a reward of $-1$ or $1$ is only recorded to the loser or winner of a match [5]. AZ is optimized to minimize the mean square error of the value estimate and reward, $v$ and $z$, and the cross entropy loss of the policy outputs of the deep network and MCTS, $\mathbf{p}$ and $\boldsymbol{\pi}$:

$$l = (z - v)^2 - \boldsymbol{\pi}^T \log \mathbf{p} + c||\theta||^2$$

MCTS balances exploration, search, and value estimates to pick the action that leads to the highest probability of winning. The loss does not intrude any term (or discount factor) to encourage winning more quickly. Consequently, the value estimates and action probabilities produced by the deep network and MCTS solely consider the actions that will eventually result in a win, *regardless of the number of moves necessary to achieve that win*. This is in contrast to similar Hex-playing agents such as MoHex [4, 8, 10], which is hard-coded to connect the winning player's pieces [1].

MoHex has been demonstrated to play perfectly on boards up to 9x9, and the AZ agent we utilize has been demonstrated to play competitively with MoHex [5]. We observe, however, that AZ does not always end the game in the fewest possible moves.

In practice, AZ sometimes delays winning. Figure 3 provides a hand-derived example of AZ delaying the end of a game. Black has nearly won the game, and must play its next move. By playing B, it can end the game; because B forms the shortest connection for black, MoHex is hard-coded to select B. When AZ's MCTS is run 100 times on this board, however, AZ places higher probability on selecting A a majority of those games (Figure 5). While playing A is a perfectly fine move, as it forms a bridge with cells B and C, it unnecessarily extends the length of the game.

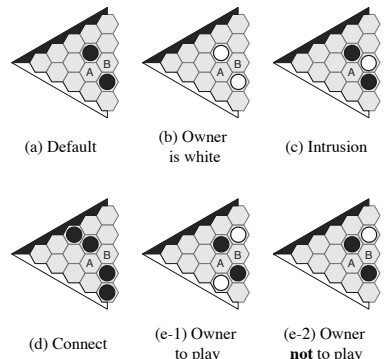

Figure 2: **Conditions.** We generate boards from all combinations of these conditions. Only "connect" (d) makes a significant impact on the results.

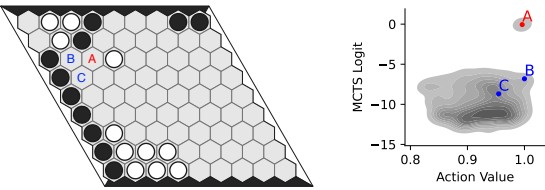

Figure 3: **Example end game board to test efficiency of last moves.** Black is close to winning the game and must decide its next move. Selecting the B ends the game, but AlphaZero instead selects A. While selecting A gives AlphaZero the ability to win the game on the next round, it unnecessarily extends the game. Right: While action B leads to a value of 1, action A also has a high value. See Appendix Fig. 5 for the per action MCTS logits.

When AZ's value estimates are all high, it often takes actions that unnecessarily extend the length of the game. In Fig. 4, we present an additional example of AZ's behavior during an endgame from a selfplay rollout. AZ has three bridges (Figure 1a) that will allow it to win the game: one with the top edge, one in the middle of the board, and one with the bottom edge. Thus, AZ is virtually connected to both edges, and guaranteed a win. However, all possible moves for black result in value estimates greater than 0.99, leading AlphaZero's MCTS to not produce significantly higher logit scores for the moves that lead to the win with fewest moves. The flatness of these value estimates suggests a lack of distinction between efficient and inefficient paths to victory.

This phenomena may have also occurred in AlphaGo's match against Lee Sedol. In game two, AlphaGo, the predecessor to AlphaZero (AZ), played a move the commentators deemed "slack" [7]. This occurred towards the end of the game (move 167 out of 211). Given that AlphaGo (and AZ) only consider the probability of winning, not the margin (nor the timeframe), again, it seems that it will sometimes waste moves when it it believes the game to be won. So, was move 167 a mistake? Was playing A in Fig. 3 or G in Fig. 4 a mistake? They are not (necessarily) mistakes, because AZ can still win. AZ may understand it can use the relevant concepts to win the game, and may eventually do so. This has ramifications for how we test AZ's behavioral understanding of concepts. To determine if AZ uses a concept, we present it with a situation where understanding the concept and using it is the only way to win.

Because AZ does not prefer faster routes, as described in the main paper, we test the agents in forced situations. In Fig. 6, not connecting the defender – i.e., testing the agent in unforced situations, the passing rate of the behavioral test is lower. This is because the agent takes a longer route to victory, not necessarily because the agent does not evaluate the concept/correct-behavior as winning.

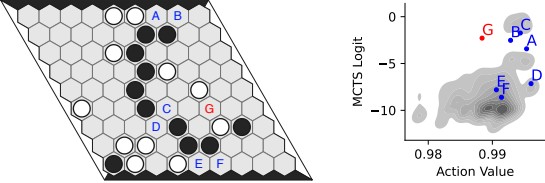

Figure 4: **Additional example endgame from selfplay.** There are six actions that move towards ending the game for black, (A-F). Sometimes, AZ instead plays G, which doesn't meaningful impact the game state. Right: Most the action values are above 0.98. Thus, the values have less impact on the MCTS logits. In 49 of 100 selfplay playouts continued from this position, AZ takes more actions than necessary. See App Fig. 5 for the per action MCTS logits.

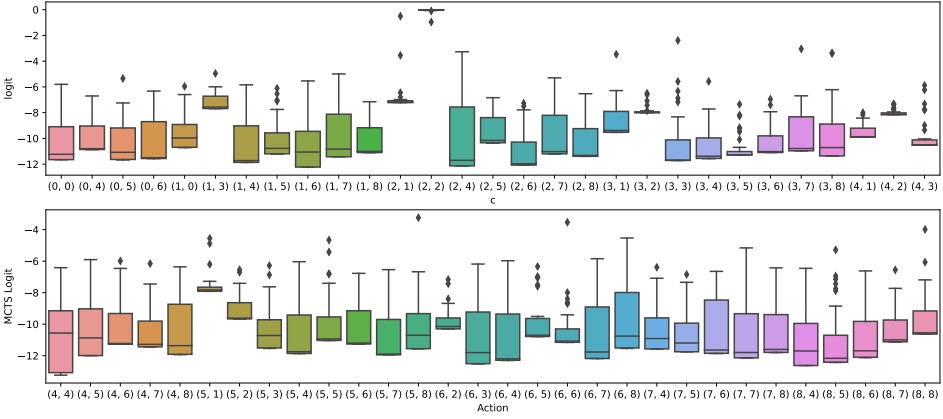

Figure 5: **Boxplot of MCTS logit values of example board presented in Figure 3.** While move 19 has high probability of being selected as the action for this board, AlphaZero places higher probability on move 20. Move 19 results in an immediate win, while move 20 results in a win in the following round of gameplay.

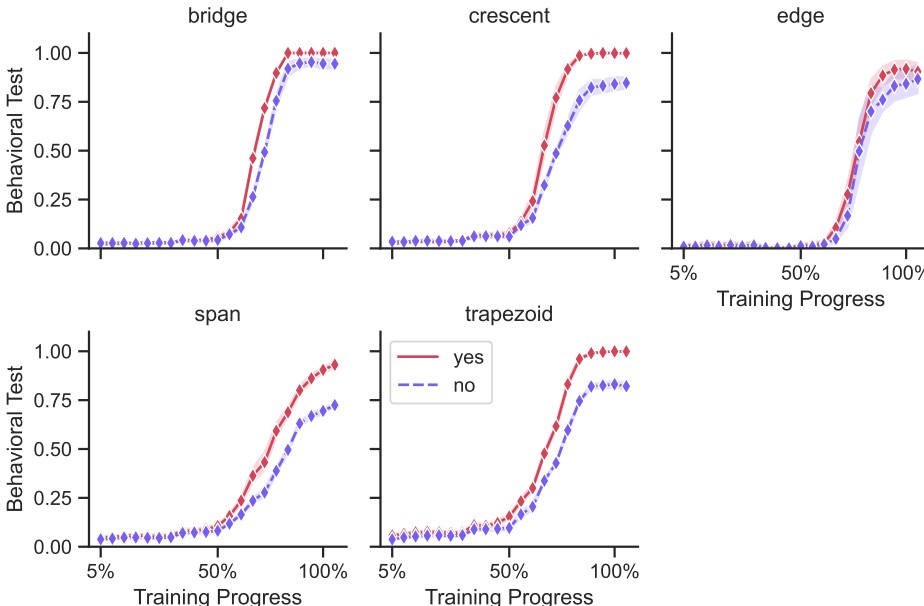

Figure 6: **Comparing behavioral results for connected vs. not connected defenders.** The passing results are lower than the connected instance because the model does not always choose the fastest route to victory.

# D   Additional Results

Fig. 7 reports key figures from the main paper replicated across architectures. Fig. 8 shows the same board structure from the main body of the paper, along with some of the quantative measures of the board structure. Fig. 9, Fig. 10, Fig. 11, Fig. 12 report the behavioral test scores (as well as "elo").

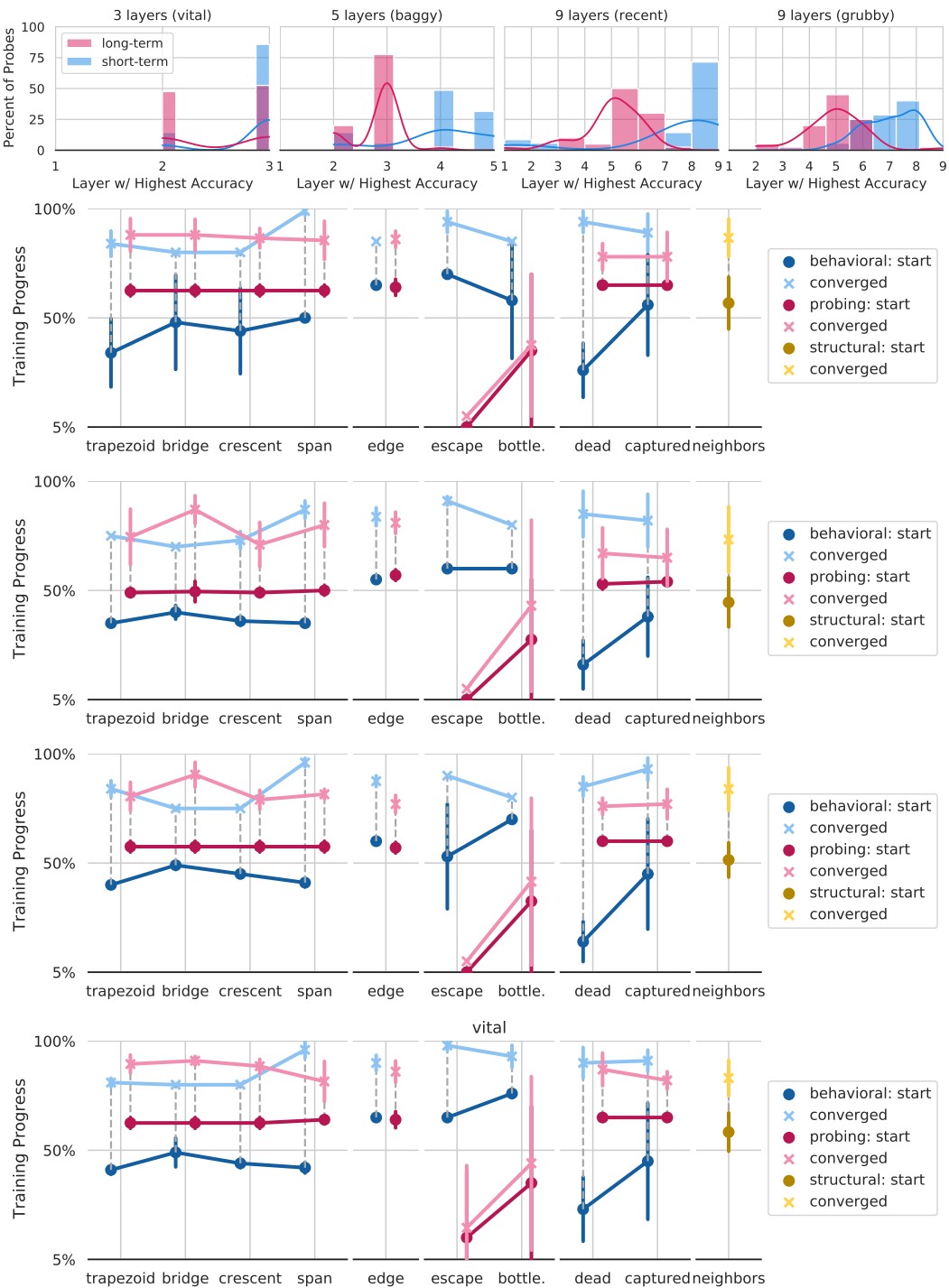

Figure 7: **Key figures from the main paper replicated across architectures.**

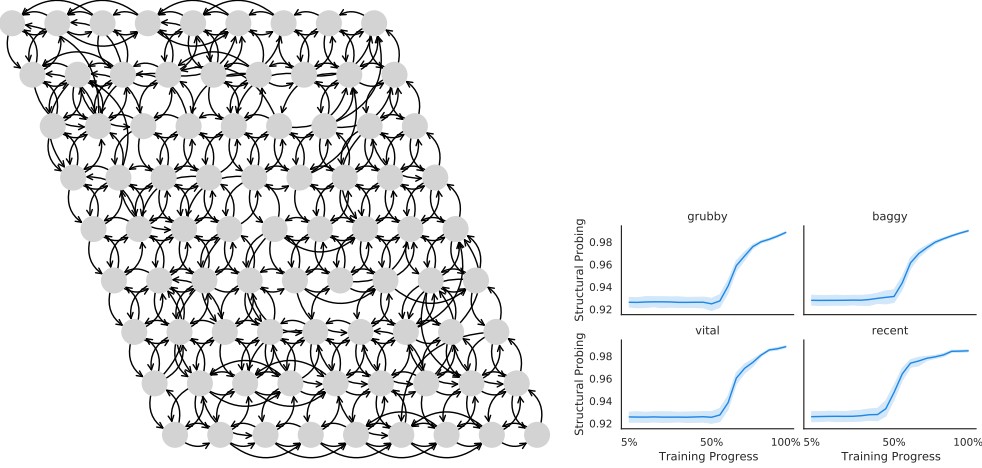

Figure 8: **Left: Dot-product scores between cell embeddings recover board structure, with modifications on the upper and lower edges.** Each grey circle corresponds to the same cell on the Hexboard; the black arrows correspond to its nearest neighbors according to dot-product scores between cells' embeddings. This diagram shows the neighborhood structure for the final checkpoint in training. See the evolution of the board structure here: `https://drive.google.com/file/d/1UV6mhnJ_FJOP3fEiHITpObUDC7lUMJx9/view?usp=sharing`. **Right: Cell embeddings encode the board structure of the game only 50% of the way through training.** We measure how well dot-product scores between cells align with the ground truth cell distance using a ranking metric Normalized Discounted Cumulative Gain (NDCG).

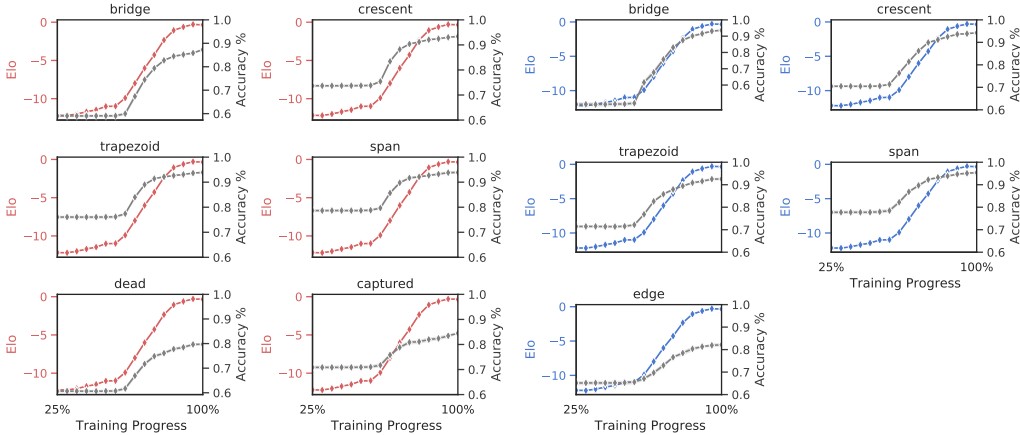

Figure 9: **Improvements in gameplay ability, as measured by Elo, coincide with improvements in concept recognition, as measured by the test accuracy of the linear probe on the highest performing layer.**

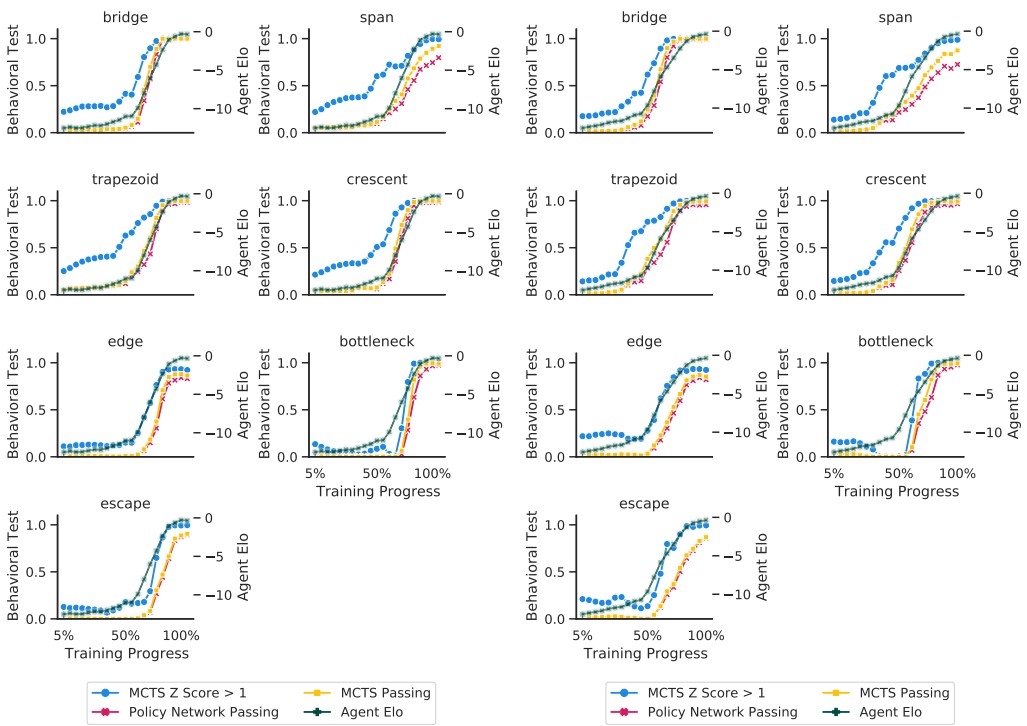

Figure 10: **AlphaZero learns to use the positive concepts; Model Code = (left)** `grubby`**, (right)** `recent`. See Table 1 for architecture details of the models. At each checkpoint, we present AlphaZero with a set of example boards that test its ability to utilize each concept. MCTS (yellow) and the deep policy network (red) select actions that pass our behavioral tests with increasing frequency throughout training. We additionally report the rate at which the action that passes our behavioral test is one standard deviation above the mean (z score > 1). The Agent Elo (dark green) measures AZ's general gameplaying ability; it increases as AlphaZero starts to use the concepts.

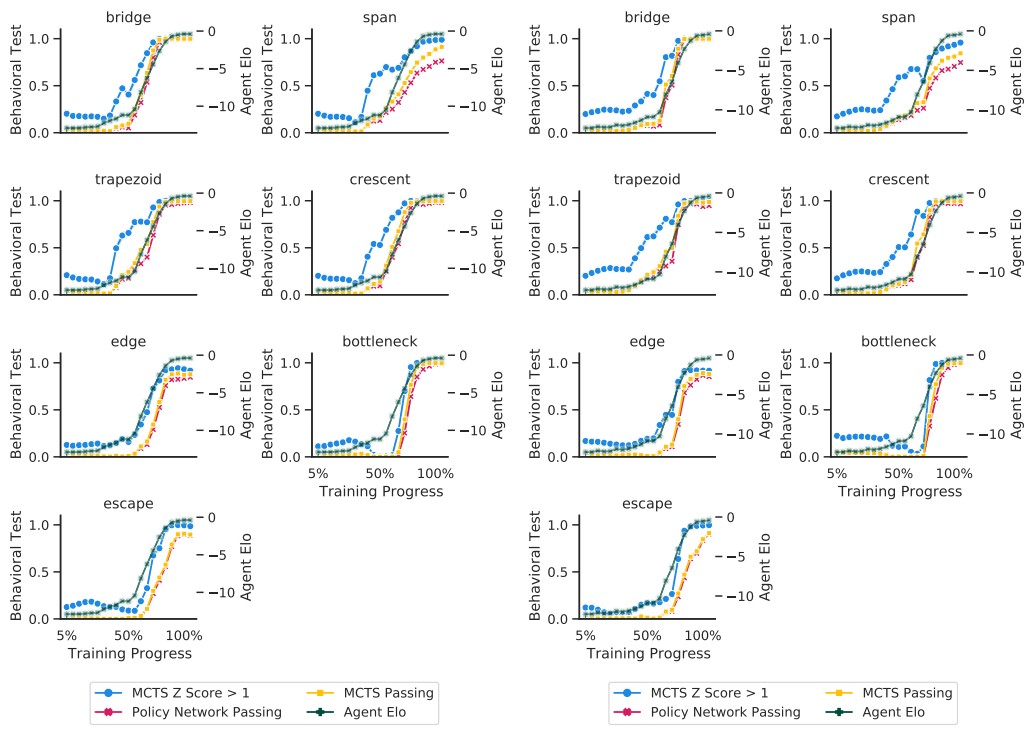

Figure 11: **AlphaZero learns to use the positive concepts; Model Code = (left)** `baggy`**, (right)** `vital`. See Table 1 for architecture details of the models. At each checkpoint, we present AlphaZero with a set of example boards that test its ability to utilize each concept. MCTS (yellow) and the deep policy network (red) select actions that pass our behavioral tests with increasing frequency throughout training. We additionally report the rate at which the action that passes our behavioral test is one standard deviation above the mean (z score > 1). The Agent Elo (dark green) measures AZ's general gameplaying ability; it increases as AlphaZero starts to use the concepts.

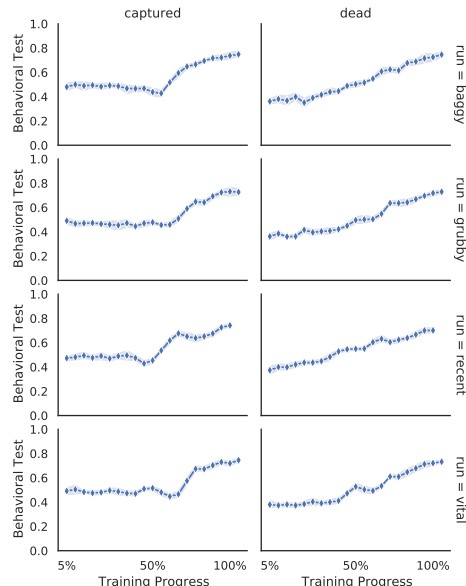

Figure 12: **AlphaZero does not fully "use" the negative concepts.** At the end of training, it plays moves in 25% of our behavioral tests that cannot impact the outcome of the game [2]. To pass these behavioral tests, AlphaZero must *avoid* playing cells on the board associated with the dead and captured concepts throughout a full selfplay rollout.