# OpenReview forum: "Evaluation beyond Task Performance: Analyzing Concepts in AlphaZero in Hex"
_NeurIPS.cc/2022/Conference — NeurIPS 2022 Accept_

### Official Review · Reviewer_CKwV · 2022-07-09

**Rating:** 6
**Confidence:** 4
**Soundness:** 2 fair
**Presentation:** 3 good
**Contribution:** 2 fair

**Summary:**

In this paper, the authors use *probing* and *behavioral* tests to investigate to what extent, and when, AlphaZero-style DNNs learn to use various concepts that humans are also known to use in the board game Hex.

The analysis shows that the DNN tends to consistently pass many of the tests, for almost all of the evaluated concepts, after about 75% training progress. Negative concepts (which recognise cells that are useless to play in, such as dead cells) are an exception; these appear to still not be fully "understood" even at the end of training, and this can be observed in the agent's behaviour as well where it sometimes does not win as fast as it could, wasting some moves instead. For most concepts, the agent's performance on behavioral tests tends to improve before the performance on probing tests, which suggests that the full MCTS agent may learn how to use them before this knowledge gets encoded into the DNN. In general, it appears that long-term concepts tend to be encoded in middle layers, whereas short-term concepts tend to be encoded closer to the final layers.

The overall methodology used for the analysis described above may be applicable to other games / other RL environments.

**Questions:**

(1) I'd like to start with a general question if you have any comments or clarifications on the weaknesses I listed above?

(2) For the probing dataset, it is described that random enemy pieces are added to the board. How many? There is no mention of randomly adding friendly pieces. Is it always guaranteed that ultimately the resulting game states constitute positions that could have been obtained through legal gameplay?

**Limitations:**

- It may be useful to discuss the apparent limitation that quite a bit of Hex-specific knowledge is used for setting up the probes (discussed in more detail as a weakness above).
- It may be useful to discuss the potential limitation I discussed in more detail above that the behavioral tests may simply all be testing for an agent's ability to recognise when it needs to interrupt an opponent's immediate winning threat.

**Strengths And Weaknesses:**

Strengths:
1. Interesting and important topic, with potential applications to improve our understanding and ability to interpret what deep RL models have learned (and when they learn it, and what they fail to learn).
2. Well-written, largely easy to follow.
3. Discussions of related work seems good, don't see anything missing there, though my background and awareness of existing work is more so in RL and games than in probing/interpretability angle.

Weaknesses:
1. I believe that one small (but important) part of the paper could use some clarifications in the writing: Section 3.2 (on Representational Probing). I will elaborate below.
2. I think that a couple of claims in the paper may be slightly too strong and need a bit more nuance. I will elaborate below.
3. A lot of the details described in Section 3.3 (Behavioral Tests) seem quite specific to the game of Hex. For the specific case of Hex, we can indeed know how to create such states that (i) contain a concept, (ii) contain that concept in only exactly one place, (iii) make sure that the agent **must** play according to the concept immediately, because otherwise it would lose. I imagine that setting up such specific situations may be much more difficult in many other games or RL environments, and would certainly require highly game-specific knowledge again for such tasks. This seems like a potential limitation (which doesn't seem to be discussed yet).

---

**On weakness (1)**:

The first sentence that I personally found confusing was "To form the random control, for each board $(H^{(0)}, y)$ in the probing dataset, we consistently map each cell in that board to a random cell, forming $H_s^{(0)}." I guess that "map each cell in that board to a random cell" means creating a random mapping from all cells in the original board to all cells in the control board, in a way such that every original cell maps to exactly one randomly-selected control cell, and every control cell is also mapped to by exactly one original cell. And then, the *value* of each original cell (black/white/empty) is assigned to the control cell that it maps to. I guess this is what is done, and it makes sense, but it's not 100% explicit. I'm afraid that many readers could misunderstand it as simply saying that every cell gets a random value directly.

Then, a bit further down under Implementation Details, it is described how the boards in the probing dataset get constructed. I suspect it would make more sense to actually describe this *before* describing how the matching controls are created.

---

**On weakness (2)**:

(a) The behavioral tests involve states created specifically such that they (i) contain the concept, but also (ii) **demand** that the agent **immediately** plays according to the concept, because it will lose otherwise. In the game of Hex, this means that all of these board states, for all these different concepts, actually include one more new "concept" that is shared across all the tests; a concept that recognises a long chain of enemy pieces that is about to become a winning connection if not interrupted by playing in what is usually just one or two remaining blank cells in between. So, I do not believe that we can say with 100% certainty that all these behavior tests are actually testing for the concept that you intend them to test for. Some or all of them may simply be testing more generally if the agent can recognise when it needs to interrupt the opponent's soon-to-be-winning chain.

(b) "Fig. 5 shows evidence that some information is learned before the model is able to use the concepts." --> I think "evidence" may be too strong here, and would say something more like "Fig. 5 **suggests** that some information **may** be learned [...]". Technically, Fig. 5 just shows that there is generally a long period with no progress on the tests, and after a long time suddenly rapid progress on the tests. To me this indeed suggests that it is *likely* that it is learning something else first, but it is not hard evidence. It could also be that it's just randomly wandering about the parameter space and suddenly gets lucky and makes quick progress then, having learned nothing at all before.

(c) "Behavioral tests can also expose heuristics the model may be using." --> yes, but only if we actually already know that the heuristics exist, and know how to explicitly encode them and create probes for them. They can't teach us any new heuristics that we didn't already know about. So maybe, better phrasing could be something like "Behavioral tests can also confirm whether or not the model may be using certain heuristics."

---

> ### Author Response · Authors · 2022-08-02
> **Thanks for your detailed review.**
>
> We address your questions and concerns as noted in your review.
>
> > The behavioral tests involve states created specifically such that they (i) contain the concept, but also
> (ii) demand that the agent immediately plays according to the concept, because it will lose otherwise.
>
> **We discuss this concern in detail in our post to all reviewers.** We will also run an additional set of experiments where the agent is not “forced” for the camera ready.
>
> > For the probing dataset, it is described that random enemy pieces are added to the board. How many?...
>
> When creating the probing dataset **we add as many pieces as needed – for both/either player – to create a valid board** that could have been obtained through legal gameplay. This can be seen in Figure 3. We will make this more explicit in the paper.
>
> > The first sentence that I personally found confusing was "To form the random control…
>
> Thanks for pointing out this point of potential confusion. **We will do as you suggest**, making our approach explicit, as well as re-order some of the content. Your understanding of what we did is correct.
>
> > I think "evidence" may be too strong here, and would say something more like "Fig. 5 suggests that some information may be learned [...]".
>
> **We understand your point and have no problem adjusting the language.**
>
> > "Behavioral tests can also expose heuristics the model may be using." --> yes, but only if we actually already know that the heuristics exist, and know how to explicitly encode them and create probes for them.
>
> This clarification that only heuristics searched for can be detected is important, and **we will add it to the paper**.
>
> > It may be useful to discuss the apparent limitation that quite a bit of Hex-specific knowledge is used for setting up the probes (discussed in more detail as a weakness above).
>
> This limitation extends to most explainability work in NLP/CV/ and recently RL. Some approaches that avoid this problem introduce their own issues, often making using the method more difficult. For example, saliency maps from CV can be helpful when trying to understand a model’s decisions, but are completely wrought with biases.  **We will update our limitations section to include this concern.**

---

### Official Review · Reviewer_8eDq · 2022-07-10

**Rating:** 5
**Confidence:** 3
**Soundness:** 3 good
**Presentation:** 3 good
**Contribution:** 2 fair

**Summary:**

The authors use linear probing to identify pre-defined Hex concepts in the activations of the agent. They use behavioural tests to evaluate whether the concepts that were identified with linear probing are actually enacted by the agent. They identify that AlphaZero does indeed represent concepts, but inconsistently enacts some of them (negative ones - the concepts related to which moves the agent should not play). They find that short term concepts (those related to winning the game) are represented in later layers in the agent and long term concepts (those related to gamestates that are distant from a potential winning state) are represented in middle layers. They also find that the learned embeddings reflect the structure of the game board.

**Questions:**

Why did the authors choose to study the game Hex? How many people play this game? Was the choice because there are known concepts for perfect play? Wouldn't chess or Go have been a more interesting game to study, given its much wider player base and the possibility to identify new concepts learned by superhuman players? The reasons behind this decision should be explained in the paper.

**Limitations:**

The authors do little to address the limitations of the method of linear probing and behavioural studies - namely, that it relies on a bank of pre-identified human concepts, consequently making it difficult to understand how the network plays so well beyond concepts humans already know. Nor does it really give us a deep picture about how the network actually achieves these feats of play. The authors adequately address the potential societal impact of their work.

**Strengths And Weaknesses:**

I'm inclined to believe the authors when they describe their work as concurrent to the (very similar) work of McGrath et al. (2021), which means the author's work is at least somewhat novel. However, the authors sell this work as borrowing methods from NLP, when in fact those methods are linear probes and behavioural studies, methods that are sufficiently general that it can't be accurate to describe them as NLP methods (even the reference the authors use for linear probing is from image classifiers; Alain and Bengio, 2017). Consequently, the discussion in the introduction regarding NLP ought to be changed to reduce the implied links between NLP and the present work. The experiments are also not especially extensive and do little to address the weaknesses of prior work, particularly those of McGrath et al., which is that they rely on a pre-defined set of concepts. Nor is it possible (in Hex) to use this method to improve human play, since perfect play for Hex is known, at least for boards of certain sizes. It would have been interesting and novel to expand on prior work to see whether AlphaZero uses novel concepts in settings where perfect play is unknown.

Overall the paper is reasonably well presented. However, figure 6 is unclear. Why are there lines connecting concepts in the same group? The authors should use a consistent name for 'structural'/'neighbours' in both the figure, its caption, and the text.

The choice of the game of Hex warrants some explanation, especially given that it dramatically reduces the audience of people who are interested in the game compared with Go or Chess.

Structural comment: Put long term vs short term (sec 2.1) into the concept taxonomy (sec 2.2)

---

> ### Author Response · Authors · 2022-08-02
> **Thank you for your review and your suggestions.**
>
> We address your larger concerns (why hex) in the response addressed to all reviewers. We address your other concerns below.
>
> > The authors sell this work as borrowing methods from NLP, when in fact those methods are linear probes and behavioural studies, methods that are sufficiently general that it can't be accurate to describe them as NLP methods (even the reference the authors use for linear probing is from image classifiers; Alain and Bengio, 2017). Consequently, the discussion in the introduction regarding NLP ought to be changed to reduce the implied links between NLP and the present work.
>
> **We can update our language to soften the implication that the methods we use are unique to NLP.**  At the same time, we would like to note that the NLP community has specifically built a sub-field of inquiry utilizing linear probes and behavioral tests to understand how well language models acquire and understand natural language concepts.
>
> > It would have been interesting and novel to expand on prior work to see whether AlphaZero uses novel concepts in settings where perfect play is unknown.
>
> **We agree that finding novel concepts and strategies learned by AlphaZero is exciting, and we believe that understanding how AlphaZero learns well-studied concepts will empower researchers to achieve this goal.**
>
> > Structural comment: Put long term vs short term (sec 2.1) into the concept taxonomy (sec 2.2)
>
> **We will update the structure of the paper.**
>
> > However, figure 6 is unclear. Why are there lines connecting concepts in the same group?
>
> **We connect the concepts within a group to highlight trends (or the lack thereof).** For example, we can see for internal concepts (bridge, span, etc.) similar behavior across concepts where the behavioral tests first improve before the probing. However, we see that the ladder concepts (escapes, bottlenecks) are quite different.
>
> > The authors should use a consistent name for 'structural'/'neighbours' in both the figure, its caption, and the text.
>
> We will use a consistent name in the paper. The motivation for doing as we had done was to highlight the purpose of the concept (structural) and the manner it was tested (neighbors). **We will use only “neighbors”.**

---

### Official Review · Reviewer_gsoX · 2022-07-11

**Rating:** 3
**Confidence:** 3
**Soundness:** 3 good
**Presentation:** 2 fair
**Contribution:** 2 fair

**Summary:**

The paper investigates AlphaZero's internal representations in the game of Hex by using two evaluation techniques: model probing and behavioral tests. They use probing classifiers to determine if agents can encode conceptual information and design behavioral tests to measure whether the agents can use the concepts to win games.

**Questions:**

* The design of behavior tests in this paper is too simple. In the experiment, the test problems are designed to be 1 or 2 moves to win and lose if playing on the wrong move. It is obvious that AlphaZero can play on the right move since it has already achieved superhuman levels. That is to say, regardless of whether the AlphaZero agents have learned the so-called "concept", they will always play on the right move by choosing the move with the highest win rate. Overall, the behavioral test experiment is not convincing and does not clarify whether AlphaZero utilizes the learned concept to do things.

* In Figure 4 (a). In [2], selectivity was used to check the complexity of the probing model, to prevent the probing model from being too complex and overfitting. However, in this paper, there is no probing model complexity comparison, only the linear classifier was used. The author should explain what is the meaning of selective in the paper.

* In line 213, the paper mentions that they have similar results with McGrath et al. [1].  (short-term was best represented higher in the network and long-term was largely represented lower in the network.)  According to [1], there is no significant evidence that short-term concepts (McGrath et al. [1]’s Fig. 2)[c,e,f] are better represented in higher layers.

Other minor issues:
* line 192, 193, what are the definitions for Logit,  z-score?
* In Figure 6, what do “behavioral:start” and “probing:start” mean? The definitions of each color point are unclear.
* In Supplementary line 34,  what A,B,C refer to?
* In Supplementary Figure7, NCDG -> NDCG

[1] T. McGrath, A. Kapishnikov, N. Tomašev, A. Pearce, D. Hassabis, B. Kim, U. Paquet, and V. Kramnik. Acquisition of chess knowledge in alphazero. arXiv preprint arXiv:2111.09259,2021.
[2] J. Hewitt and P. Liang. Designing and interpreting probes with control tasks. InProceedings of the 2019 Conference on Empirical Methods in Natural Language Processing and the 9th International Joint Conference on Natural Language Processing (EMNLP-IJCNLP), pages 2733–2743

**Limitations:**

The authors have addressed most of the possible limitations and potential negative societal impact in Section 3.4.

**Strengths And Weaknesses:**

The paper combines both representational and behavioral approaches to analyze reinforcement learning agents in the game of Hex. Overall, the result is quite interesting and allows the readers an understanding of how AlphaZero agents learn. However, the method used in the paper has already existed and has been widely used in many other board games, such as Chess[1], and Go[2]. The paper simply applies these methods to a different game (Hex). I think the novelty is not enough and the results in this paper will be limited to the whole community.

[1] T. McGrath, A. Kapishnikov, N. Tomašev, A. Pearce, D. Hassabis, B. Kim, U. Paquet, and V. Kramnik. Acquisition of chess knowledge in alphazero. arXiv preprint arXiv:2111.09259,2021.
[2] N. Tomlin et al. Understanding game-playing agents with natural language annotations. To Appear: ACL, 2022.

---

> ### Author Response · Authors · 2022-08-02
> **Thank you for your detailed review!**
>
> Thank you for your detailed review. We worked to address some of the issues you raised above in the post for all the reviewers. In particular, we restated why we designed the behavioral tests as we did. In this post, we will go over how our work relates to McGrath et al and Tomlin et al. In the following posts we will address your other questions.
>
> **Our work is concurrent to Tomlin et al. and McGrath et al. Our findings also differ from McGrath et al. and Tomlin et al. in three notable ways:**
>
> 1. We analyze the role of MCTS in learning concepts using behavioral tests and demonstrate that **MCTS is the means by which concepts are first identified and encoded into the network.** Additionally, we show that concepts encoded by AlphaZero can be utilized in gameplay.
> 2. We organize board concepts into hierarchies, which allow us to compare concepts across board conditions. **Concepts are learned at different layers of the network depending on the state of gameplay (long vs short), and concepts that are denoted by avoiding moves, “negative concepts”, remain difficult to learn by AlphaZero.**
> 3. **We consider alternative training architectures of AlphaZero and are able to confirm that our results apply to other AlphaZero agents trained on Hex.**
>
> While we provide a qualitative evaluation of the results presented by the authors of concurrent work in their respective paper, we do not believe that strict comparisons of our work to McGrath et al. are reasonable as McGrath et al. has yet to be published in a peer reviewed venue and relies on a computationally expensive model and closed-source codebase.

---

> > ### Author Response · Authors · 2022-08-02
> > **Smaller issues.**
> >
> > Thank you for pointing out the smaller issues you saw with the paper as well. Below we address the minor details you raised. We will update the paper accordingly.
> >
> > > line 192, 193, what are the definitions for Logit, z-score?
> >
> > We will clarify the definitions in 192/193 and include a clear procedure in the supplementary material. **The logits are the outputs of the given module (MCTS or policy prediction).** These logits can be transformed into a probability distribution over actions. A Z-score captures how “different”, in our case how much higher, a given value is compared to the mean value in a population. In particular, **the Z-score reports the number of standard deviations a value is above the mean.** Here, the population are the logit scores over the actions. Thus, as the Z-scores of the correct actions increase, we can understand that as meaning the correct actions are becoming more likely.
> >
> > > In Figure 6, what do “behavioral:start” and “probing:start” mean? The definitions of each color point are unclear.
> >
> > We apologize for the confusion here; we will clarify this in the paper. In this chart **we aim to report when the agents start to improve on the probing/behavioral-tests, and when their performance converges.** We denote when the agent starts to improve as “...:start” and we denote converging as “...:converged.” In particular, we report these as when there is an improvement of 5% over the baseline value.
> >
> > > In Supplementary line 34, what A,B,C refer to?
> >
> > Thanks for catching this! B,C had once referred to the cells directly below A. **We will fix this error.**
> >
> > > In Supplementary Figure7, NCDG -> NDCG
> >
> > Thanks for catching this! **We will fix this error as well.**

---

> > ### Author Response · Authors · 2022-08-02
> > **Some of your remaining concerns.**
> >
> > > The design of behavior tests in this paper is too simple. In the experiment, the test problems are designed to be 1 or 2 moves to win and lose if playing on the wrong move…
> >
> > Our tests are limited to concepts with specific sets of implied moves and strategies. However, **knowing when these AlphaZero agents fail these tests help us understand when and how these strategic capabilities are learned.** We clarify the motivation for our design decisions for these tests (and the tradeoffs) in our post directed to all the reviewers.
> >
> > > In Figure 4 (a). In [2], selectivity was used to check the complexity of the probing model, to prevent the probing model from being too complex and overfitting. However, in this paper, there is no probing model complexity comparison, only the linear classifier was used. The author should explain what is the meaning of selective in the paper.
> >
> > **In both our work and in Hewitt et al.’s paper the meaning of selectivity is the same: The performance of a probe on a given task beyond the performance of a probe on a control task.** Hewitt et al. used this to help design and select among different probing architectures, but also noted that selectivity is important to understanding a given probe’s performance. It adds context for interpreting the results. For example, the probing results for escapes and bottlenecks report a high accuracy and low selectivity. This suggests that the representation isn’t key to the high accuracy performance. This provides context to the results regarding when different concepts are first learned (Fig. 6).
> >
> > > In line 213, the paper mentions that they have similar results with McGrath et al. [1]. (short-term was best represented higher in the network and long-term was largely represented lower in the network.) According to [1], there is no significant evidence that short-term concepts (McGrath et al. [1]’s Fig. 2)[c,e,f] are better represented in higher layers.
> >
> > The authors of [1] did not seem to note the distinction between short- and long-term concepts. However, when examining their results we found that a similar pattern emerges. **This observation is our interpretation of the results shown in their plots and not a direct paraphrasing of their discussion.**

---

### Official Review · Reviewer_YogY · 2022-07-12

**Rating:** 6
**Confidence:** 4
**Soundness:** 3 good
**Presentation:** 2 fair
**Contribution:** 3 good

**Summary:**

This paper studies models trained with AlphaZero for the game of Hex. The paper applies techniques from the NLP literature to investigate whether the learned model learns concepts usually taught to human players. The paper further investigates if the trained agent is able to act according to such concepts (e.g., connect a bridge when that is needed to win the game or to reach an advantageous state of the game).

**Questions:**

Why did you define short term as single moves to the end of the game and more than one move as long term? It would be interesting to see a continuum between short and long term. For example, one could argue that 2 moves to the end of the game is still short term.

Why is B a captured cell in Figure 2 (j)? This should be explained in the main text.

Is Hˆ{(l)} a board or the activation values of the network? I would have appreciated a formal definition of weights and activation values as I am not sure I would be able to implement the Probe classifier with the current description from the paper. For example, do we use the output values of all neurons of all layers as input to the classifier? Do we use the logits or the values after activation?

**Limitations:**

The experiments were performed on a single domain and it isn't clear whether they will generalize to other domains (e.g., security games or even other board games).

The experiments and results aren't exactly actionable. They are of scientific interest and might inspire others, but it isn't clear what role this paper will have moving forward.

**Strengths And Weaknesses:**

Strengths

The paper is well written and the experiments reported are interesting. Although the metrics and procedures come from the NLP literature, this paper is the first to attempt them in models learned with AlphaZero. I find it particularly interesting the Probe procedure, where a linear classifier is trained to detect known concepts of Hex. Interestingly, the concepts studied are detected with very high accuracy and the accuracy improves as AlphaZero further trains the model. This shows an interesting connection between the learned model and concepts that we (humans) deem as important for playing the game.

There is a chance this paper will inspire others to work on the extraction of such concepts from the learned model. This future research endeavor could be important for those interested in teaching humans the knowledge that is generated with learning systems.

The paper also makes interesting connections with previous work. I particularly like the connection between the results presented in the paper and those of McGrath et al. [37] on chess regarding where in the neural model that the short and long-term decisions are stored. Such connections facilitate the work of others starting out in this field.

It was also interesting to see the methodology the paper uses to verify that the learned agent learns the connections in the board of Hex.

Weaknesses

Some of the results reported in the paper aren't as interesting as the idea of running the experiments. For example, short-term decisions are encoded near the output layers of the neural model. Perhaps this isn't surprising because such decisions are actionable and one would expect the last year of the model (which is also linear) to correctly identify such situations.

Another result that perhaps isn't particularly interesting is the fact that the studied concepts are first noted in the MCTS search and only later incorporated in the model. This is because MCTS generates the training data for the model, so the natural order is to first generate the data and then train the model. It would be strange if the model learned before MCTS collected the data.

It is interesting that the concepts 'escape' and 'bottleneck' are first detected in probing to only then being detected in decisions of the agent (Figure 6 of the paper). I list this as a 'weakness' because the paper offers no explanation of why this is the case. This result almost contradicts the idea of MCTS generating training data and the model learning from the training data. This is because the concept can be first noted in the neural model to only then appearing as decisions in the game (behavior). I apologize if I missed the explanation of this phenomenon and would appreciate a response from the authors about it.

Other comments

Regarding the following: "This highlights a weakness in AlphaZero and a risk: Some concepts may be provable and useful to people, but deemed less important by AZ." The agent is probably playing dead cells because it has either won or lost the game and it doesn't matter if it plays on those cells. If this is the cases, then this isn't a weaknesses of AlphaZero, but rather a challenge for systems that will attempt to extract information from the learned model to transfer to human learners.

---

> ### Author Response · Authors · 2022-08-02
> **Thanks for your detailed review!**
>
> Thanks for your review!  We’ll address your concerns in turn:
>
> > Some of the results reported in the paper aren't as interesting as the idea of running the experiments. For example, short-term decisions are encoded near the output layers of the neural model. Perhaps this isn't surprising because such decisions are actionable and one would expect the last year of the model (which is also linear) to correctly identify such situations.
>
> We agree that the results are not necessarily "surprising". **However, whether results/hypotheses are surprising or unsurprising is largely orthogonal to what research we should conduct and value. Excluding papers from conferences/journals in this way can leave us open to biases and misunderstandings.** Just as we discourage p-hacking to favor positive results, we should encourage publishing work of “scientific interest” that happens to report unsurprising results.
>
> **We agree with your expectation: This is also our running hypothesis for why the short-term concepts are best-represented near the end of the network.** Still, it is interesting that the long-term concepts have a different pattern of results.
>
> > Another result that perhaps isn't particularly interesting is the fact that the studied concepts are first noted in the MCTS search and only later incorporated in the model. This is because MCTS generates the training data for the model, so the natural order is to first generate the data and then train the model. It would be strange if the model learned before MCTS collected the data.
>
> While your intuition makes sense and is consistent with what we observe, **it is not necessarily obvious prior to seeing the results of our experiments. Alternative possibilities include:**
>
> 1. **The network may never learn to handle the concepts because MCTS handles it for the network.** While the loss might preclude this from happening in the limit, it isn’t clear in practice what would occur.
>
> 2. **The network could have started to represent the concepts due to the value prediction, before the MCTS “collected the data.”**
>
> > Why is B a captured cell in Figure 2 (j)? This should be explained in the main text…
>
> **We will add an explanation to the main text.**
>
> **B is not a captured cell; instead, playing B makes A captured.**  The full explanations of each of the boards are presented in the supplementary on lines 37-40: “An empty cell is considered captured when it is effectively filled by a player. The templates have (at least) two empty cells A and B (Fig. 1f). If the cells are black-captured, it means that if white intrudes into the template (playing A), then black can respond by playing B and making A dead.”
>
> > Is Hˆ{(l)} a board or the activation values of the network?...
>
> **H is either the raw board representation or the activations of a single layer in AlphaZero’s network body.** $H^{(0)}$ is the raw representation of the board input to AlphaZero. $H^{(l)}$ is the activation of the lth layer of AlphaZero’s network body. As we noted earlier, the network body is before the output policy and value heads and does not include them. We never use activations from multiple layers, neuron weights, AZ’s value predictions, nor AZ’s policy prediction.
>
> $H^{(0)}$ = board
>
> $H^{(l)}$ = AlphaZero.body\[l-0\]($H^{(l-1)}$), for l in 1..L.
>
> > It is interesting that the concepts 'escape' and 'bottleneck' are first detected in probing to only then being detected in decisions of the agent (Figure 6 of the paper). I list this as a 'weakness' because the paper offers no explanation of why this is the case…
>
> This result is a particularity of escape/bottleneck. In Fig 6’s caption, we noted this exception. **Escape and board configurations occur at the edge of the board, and as a result, they have few possible configurations and are easy to detect.** We can see they are empirically easy to detect from Fig. 4(a) which shows that these two concepts have low selectivity. **We will update the paper to include this explanation, thank you for pointing that out.**
>
> > Why did you define short term as single moves to the end of the game and more than one move as long term? It would be interesting to see a continuum between short and long term. For example, one could argue that 2 moves to the end of the game is still short term.
>
> Yes, adding a continuum would be interesting. **We can introduce this as a supplementary result in the camera ready.** Note that for many of the concepts, there is more than one move until the end of the game.

---

> > ### Comment · Reviewer_YogY · 2022-08-05
> > **Thanks**
> >
> > Thank you for your responses.
> >
> > I just wanted to clarify that I am not arguing that the paper shouldn't be published because some of the results aren't surprising. First, I gave a positive score to the paper and I don't see the weaknesses I listed in my review as fatal. Second, I didn't mean to say that the results were unsurprising. I meant to write that some of the results aren't interesting in the sense that I don't know what to do with the information they provided -- I am just indifferent to what those particular experiments taught us.
> >
> > The Hˆ{(l)} notation wasn't very clear to me when I read the paper. It could be that I just missed its definition; if not, please clarify it in the camera ready if the paper is accepted.

---

> > > ### Author Response · Authors · 2022-08-08
> > > **Thank you for these clarifications**
> > >
> > > Based on your feedback, we plan in the camera ready to describe further how the insights of the paper could be used.
> > >
> > > Additionally, we will add additional clarifications to our definition and notation of $Hˆ{(l)}$ in Section 3.2 to better communicate how we build our representational probes.

---

### Author Response · Authors · 2022-08-02
**Response to All Reviewers**

**Thank you all for your thoughtful reviews.** We address each of your concerns in detail in separate posts below. In this set of posts, we answer why study Hex in particular and why we forced moves in our behavioral tests.

---

> ### Author Response · Authors · 2022-08-02
> **Forcing Moves in Behavioral Tests [gsoX, CKwV]**
>
> **We designed our behavioral tests to force moves, as otherwise, the agent may take a longer but also winning route.** We discuss this behavior by AlphaZero in the supplementary Section C, in which we describe and provide examples of how AlphaZero selects moves without consideration of the number of moves necessary to win.
>
> **Testing a model in a forced situation has been demonstrated to provide a strong signal for when an agent does not use a concept.** This approach has been used to test large language models. Specifically, we are inspired by Ettingers [3, 4] who uses simplified language to ask if the model is able to use a concept like negation or syntax when needed. If the model still fails to use the concept, then this good evidence the concept is not represented. In the same way, we use these forced behavioral tests to learn when AlphaZero does not use a concept and when it starts to use it.
>
> Forcing the moves introduces potential challenges. For example, as noted by CKwV, the concept being tested could be interpreted as “interrupt the opponent's soon-to-be-winning chain.” However, the behavioral tests of different concepts report different performance levels (though admittedly similar), suggesting that the differences in scenarios/concepts are still relevant. Even if there is a “stop-the-soon-to-be-winning-chain” concept, that does not take away from the fact that the agent had to use the targeted concepts as a constituent of “stop-the-soon-to-be-winning-chain.”
>
> **Finally, the alternative approach of not forcing moves makes it difficult to to cleanly answer the questions we are trying to ask.** Because the model could take a longer route to victory, interpretation of the results becomes murky. Simply filtering these results may create biases in the numeric results.
>
> [3] https://arxiv.org/pdf/2109.12393.pdf
>
> [4] https://aclanthology.org/2020.tacl-1.3/

---

> > ### Comment · Reviewer_CKwV · 2022-08-04
> > **On forcing moves in behavioral tests**
> >
> > Thanks for your responses to the reviews!
> >
> > I fully understand **why** the behavioral tests were set up in the way that they were (i.e., setting them up in a way such that if the agent does not immediately play according to the concept, they'll lose), but I still think it's a potential... issue, that at least warrants discussing.
> >
> > Even if there are several good reasons for choosing this setup, it still means that there is a risk that the tests are not 100% testing for the concepts that they are intended to test for. So it really does require some discussion / nuance at least.
> >
> > Note that, personally I do not view this as a major issue in terms of the contributions/significance of the work. I cannot think of a good solution that would avoid this problem myself either, we can't definitively say that the tests are *not* testing for the intended concepts either, and I think it's fine for papers to make partial progess without immediately and definitively solving all the potential problems in one go. I mostly view it as an issue that does require acknowledgement, discussion, and nuance.

---

> > > ### Author Response · Authors · 2022-08-09
> > > **Adding Additional Discussion of Limitations of Behavioral Testing Setup**
> > >
> > > Thank you for your suggestions and feedback. We absolutely agree and plan to add additional details of the limitations of our behavioral tests in our considerations section (Section 3.4).

---

> ### Author Response · Authors · 2022-08-02
> **Studying Hex [gsoX,  8eDq]**
>
> We believe using Hex as an instrument to study AlphaZero’s behavior throughout training and how internal representations and external behavior (begin to) relate is scientifically interesting. As we briefly discussed in the introduction (L44), there are a few good reasons for using Hex:
>
> 1. **Perfect-play baselines are a clear strength for using Hex, which we used to help design the behavioral tests.** Solvers that can evaluate for perfect play such as MoHex allow us to evaluate the quality of the model’s policy, because we can measure how well AlphaZero performs against the optimal strategy. Furthermore, we can (and did) use the “ground truth” moves provided by MoHex to help design our behavioral tests.
> 2. **Hex also provides practical benefits to the study of AlphaZero because it is computationally cheaper, allowing us to ask interesting questions about the learning dynamics of AlphaZero.**  AlphaZero, used in McGrath et al. is a closed-source single model. Elf OpenGo, used by Tomlin et al. is an open-source single model at a single checkpoint. As noted in Sellam et al. [1] and Henderson et al. [2], evaluating an experiment on multiple versions of a model is important to understand to what extent do results generalize. Jones trained and shared a number of AlphaZero models with their checkpoints, across varying model architectures and game conditions. **The variety of models provided by Jones at various points allow us to ask the question of how and when concepts are learned and how well these trends hold across architectures.**
>
> [1] https://arxiv.org/pdf/2106.16163.pdf
>
> [2] https://ojs.aaai.org/index.php/AAAI/article/view/11694

---

### Meta-Review · Area_Chair_jFGK · 2022-08-23

**Recommendation:** Accept
**Confidence:** Less certain

**Metareview:**

This paper uses methods from interpretability to study the knowledge learned by AlphaZero in the game of Hex. In particular, the network's outputs are correlated with various hand-designed features.

The reviewers are all in agreement that there are some interesting contributions. There was some comparison with McGrath et al., but given that this paper is relatively recent and still unpublished, this does not seem like an insurmontable blocker. There was also some debate on whether evaluating on Hex is sufficiently relevant to the research community, or whether the kind of probes used are particularly insightful. I agree that the latter is a weak point of the paper, while the former is a reasonable concern but more minor. Another concern is what we do from this paper onwards -- i.e., how will this research feed into future work? In follow-up discussion, a majority of reviewers argued that there were useful nuggets of knowledge produced by this paper.

**Award:**

No

---

### Decision · Program_Chairs · 2022-09-14

Accept